# Refining CT image analysis: Exploring adaptive fusion in U-nets for enhanced brain tissue segmentation

**Bang-Chuan Chen[1], Chung-Yi Shen[2], Jyh-Wen Chai[3,4], Ren-Hung Hwang [5], Wei-Chuan Chiang[2], Chi-Hsiang Chou [1,3,6,7]\*, Wei-Min Liu[2]\***

**1** The Department of Neurological Institute, Taichung Veterans General Hospital, Taichung, Taiwan, ROC, **2** Department of Computer Science and Information Engineering and Advanced Institute of Manufacturing with High-tech Innovations, National Chung Cheng University, Chiayi, Taiwan, ROC, **3** Department of Post-Baccalaureate Medicine, College of Medicine, National Chung Hsing University, Taichung, Taiwan, ROC, **4** Department of Radiology, Taichung Veterans General Hospital, Taichung, Taiwan, ROC, **5** College of Artificial Intelligence, National Yang Ming Chiao Tung University, Hsinchu, Taiwan, ROC, **6** Department of Neurology, School of Medicine, College of Medicine, National Yang Ming Chiao Tung University, Taipei, Taiwan, ROC, **7** Department of Neurology, School of Medicine, National Yang Ming Chiao Tung University, Taipei, Taiwan, ROC

\* bryanchouysvh@gmail.com (CHC); wmliu@cs.ccu.edu.tw (WML)

## Abstract

### Purpose

Non-contrast Computed Tomography (NCCT) quickly diagnoses acute cerebral hemorrhage or infarction. However, Deep-Learning (DL) algorithms often generate false alarms (FA) beyond the cerebral region.

### Methods

We introduce an enhanced brain tissue segmentation method for infarction lesion segmentation (ILS). This method integrates an adaptive result fusion strategy to confine the search operation within cerebral tissue, effectively reducing FAs. By leveraging fused brain masks, DL-based ILS algorithms focus on pertinent radiomic correlations. Various U-Net models underwent rigorous training, with exploration of diverse fusion strategies. Further refinement entailed applying a 9x9 Gaussian filter with unit standard deviation followed by binarization to mitigate false positives. Performance evaluation utilized Intersection over Union (IoU) and Hausdorff Distance (HD) metrics, complemented by external validation on a subset of the COCO dataset.

### Results

Our study comprised 20 ischemic stroke patients (14 males, 4 females) with an average age of 68.9 ± 11.7 years. Fusion with UNet2+ and UNet3＋yielded an IoU of 0.955 and an HD of 1.33, while fusion with U-net, UNet2＋, and UNet3＋resulted in an IoU of 0.952 and an HD of 1.61. Evaluation on the COCO dataset demonstrated

**Data availability statement:** The brain CT data cannot be shared publicly due to ethical restrictions concerning patient confidentiality and privacy, enforced by the Institutional Review Board (IRB) of Taichung Veterans General Hospital. Data requests can be sent to the IRB office at Taichung Veterans General Hospital (email: clsu@vghtc.gov.tw) for consideration of eligibility and requirements for access.

**Funding:** The work is supported by the Advanced Institute of Manufacturing with High-tech Innovations, National Chung Cheng University, Chiayi, Taiwan and Taichung Veterans General Hospital with grant number TCVGH-1085505C (to JWC). It is also supported by the National Science and Technology Council with grant numbers 110-2221-E-194 -011 (to JWC) and NSTC 113-2221-E-075A-008-MY2-1 (to CHC). The funders provided financial support for the Article Processing Charge (APC) but had no role in study design, data collection and analysis, decision to publish, or preparation of the manuscript.

**Competing interests:** The authors have declared that no competing interests exist.

an IoU of 0.463 and an HD of 584.1 for fusion with UNet2+ and UNet3+, and an IoU of 0.453 and an HD of 728.0 for fusion with U-net, UNet2+, and UNet3+.

## Conclusion

Our adaptive fusion strategy significantly diminishes FAs and enhances the training efficacy of DL-based ILS algorithms, surpassing individual U-Net models. This methodology holds promise as a versatile, data-independent approach for cerebral lesion segmentation.

---

## Introduction

Stroke, the second leading cause of global mortality, claims approximately 5.5 million lives annually [1]. It manifests as ischemic or hemorrhagic stroke. While ischemic stroke lesions are clearer on MRI, the process is time-intensive compared to CT scans. In emergencies, CT offers quicker access, reducing door-to-puncture time and enhancing brain tissue reperfusion post-treatment. Hence, prompt lesion identification remains clinicians' top priority.

Advances in artificial intelligence (AI) enable rapid and precise stroke diagnosis. Brain tissue segmentation (BTS) is crucial for diagnosing strokes and tumors. Various methods using structural MRI data have been proposed [2–6]. Liu et al. (2007) automated BTS through Diffusion Tensor Imaging space fusion [7], while Reza et al. (2013) used machine learning for abnormal BTS and classification [8]. Pacheco et al. (2023) investigated MRI-based BTS and brain tumor segmentation [9].

Otsu's method (1979) segments CT images using bimodal histograms [10], and Lauric and Frisken successfully segmented brain tissues under CT using various methods [11]. Chan and Vese proposed an active contours method in 2001 [12], and Irimia et al. (2019) achieved promising CT BTS results [13].

U-Net is widely used for organ and lesion segmentation [14]. Variations like H-DenseUNet for liver tumor segmentation (2017) [15], Multi-scale U-Net (MSU-Net) (2021) [16], and UNet++ (2018) [17] have been developed. Huang et al.'s UNet3+ (2021) addresses full-scale segmentation challenges [18]. While many BTS methods using deep learning exist [19–22], most focus on MRI data rather than CT images.

Ensemble learning is a widely used technique in machine learning, combining multiple models to boost performance and minimize errors [23]. Stacking, a key subcategory, involves using various models on the same data and leveraging another model to optimize their predictions, resulting in higher accuracy than any single model. In semantic segmentation, ensemble methods have been proven effective in enhancing performance [24–25], offering more precise segmentation compared to individual networks. Fusion operations consolidate multi-view information, improving image quality and retaining critical features [26]. Despite its benefits, ensemble learning may face challenges when results vary significantly.

Differentiating brain structures from surrounding tissues, especially outside the cortical region, poses challenges, leading to over-segmentation. We evaluated U-Net,

UNet++, and UNet3+ on an anonymized dataset, noting their tendency to over-segment surrounding soft tissues. The accompanying Fig 1 illustrates examples of BTS, demonstrating the misclassification of some non-brain and subcutaneous tissues as brain tissue, resulting in over-segmentation. A comparison of ground truth in Fig 1b and 1e reveals misclassified non-brain and subcutaneous tissues as brain tissue in Fig 1c and 1f, leading to over-segmentation. To address this, we proposed an adaptive result fusion strategy, enhancing deep-learning (DL) models' accuracy across different slice layers and facilitating stroke diagnosis.

## Materials and methods

### Devices

The major hardware and operating system used in this study are listed below:

- Central Processing Unit (CPU): two Intel Xeon Silver 4110 CPUs;

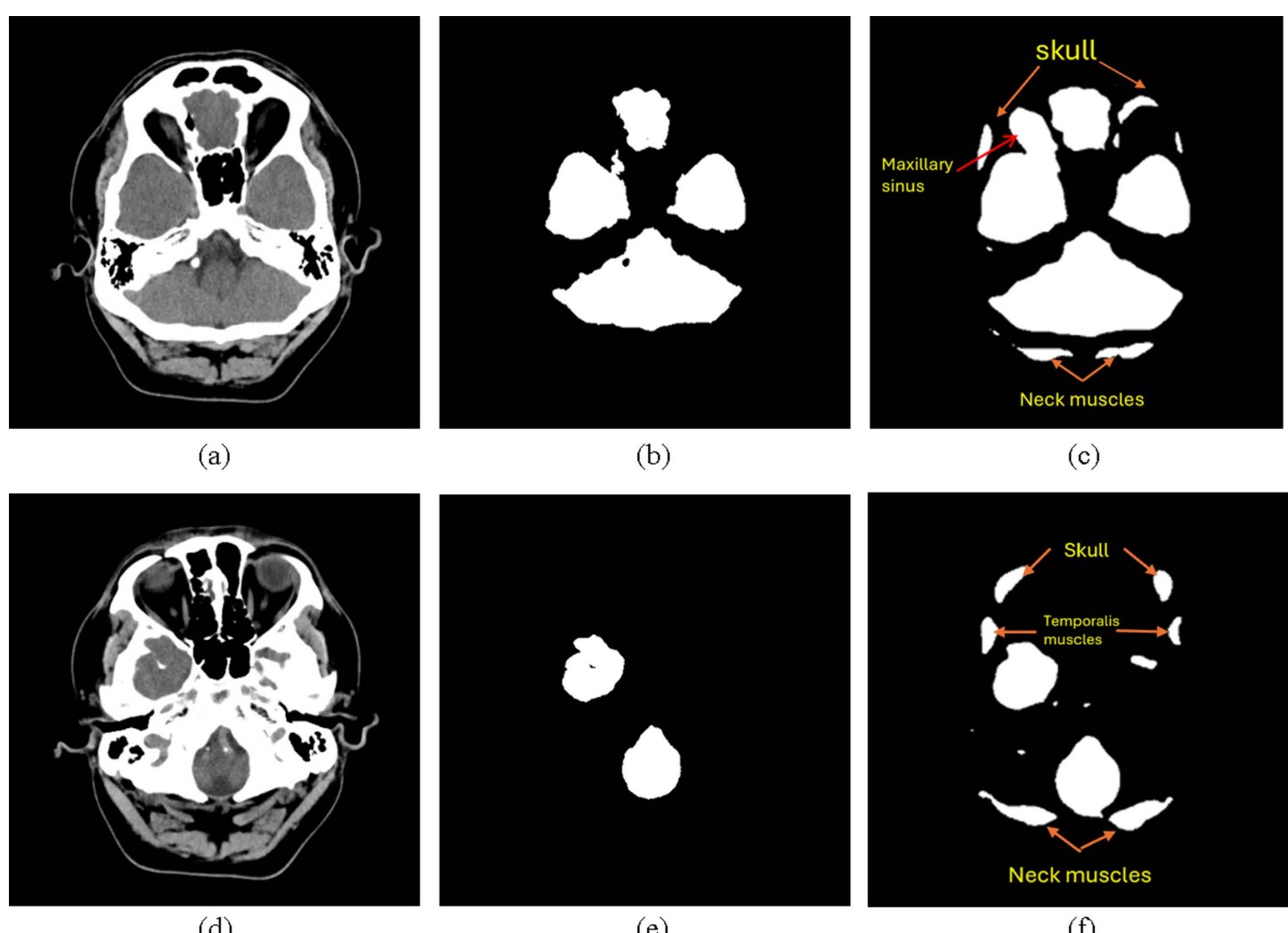

**Fig 1. Brain tissue segmentation examples.** Two slices in a whole-brain CT scan (a, d), their ground truth brain tissue masks (b, e), and the over-segmented results by some brain tissue segmentation algorithms (c, f).

- Memory: 314 gigabytes;

- Graphics Processing Unit (GPU): two Nvidia RTX 2080 Ti (11 GB);

- Operating system: 64bit Ubuntu 18.04.1 LTS;

- Software development environment: CUDA 10.2, cuDNN 7.6.5, Python 3.7.9, Pytorch 1.7.0, and openCV-python 4.1.2.

## Patient collection and CT Image acquisition

We conducted a retrospective analysis of ischemic stroke patients aged ≥ 20, discharged from Taichung Veterans General Hospital between February 2016 and November 2020, following approvals (permit number CE19010B, CE19010B-1 and CE19010B-2) from the hospital's Institutional Review Board. According to the approvals, we accessed the brain CT scan data of these patients from February 1st, 2019, until January 31st, 2022. Although the authors had access to information that could identify individual participants during or after data collection, we have implemented measures to safeguard participant confidentiality, including data de-identification and password protection.

Using a Philips Brilliance 64 CT scanner, non-contrast brain CT scans (NCCT) were obtained with imaging parameters set at 120kVp tube voltage and 250mAs tube current time. Each image series, comprising over 30 slices, covered the entire head from jaw to top, as illustrated in Fig 2 (adapted from COCO dataset [27]). Notably, lower brain tissue

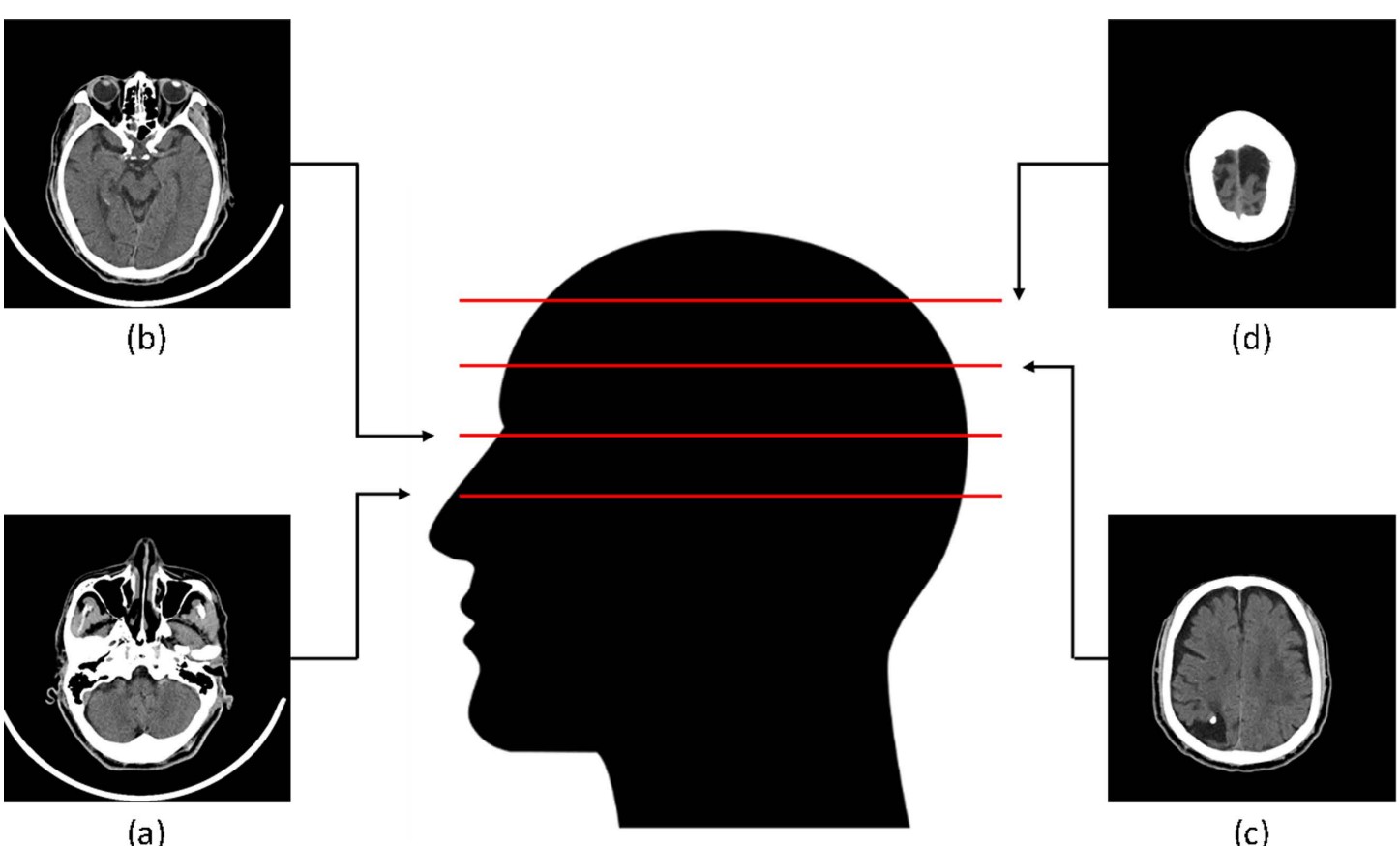

**Fig 2.** (a) The first few slices of a whole-brain CT scan cover the lower part of the brain; (b) The 3rd slice contains the eyes; (c) The 19th slice has a simple structure and is easy for brain segmentation; (d) The last slice is near the top of the head.

layers (Fig 2a and 2b) exhibited varied shapes, potentially including non-brain tissues like the tongue, eyes, and cheeks.

Brain tissue boundaries in each image were manually delineated by neurologists and radiologists, who also assisted in identifying the stroke area. This outlined area served as the ground truth mask (refer to Fig 1). In images without brain tissue, the ground truth was uniformly represented as black.

## Training

The labeled NCCT slices underwent random partitioning into training and test datasets, maintaining a 9:1 ratio, stratified by patient groups. The training process consisted of 50 epochs utilizing BCELoss as the loss function. Augmentation techniques such as random rotation, vertical flip, and horizontal flip were applied to the training dataset.

We experimented with different optimizers, including Adagrad and Adam, each with initial learning rates of 0.001 and 0.0001, respectively. The 'ReduceLROnPlateau' function in PyTorch dynamically adjusted the learning rate to ensure the convergence of model training.

To evaluate the generalization capability of our adaptive fusion strategy, we trained models for cat segmentation using an additional set of 4298 cat images from the COCO dataset. These images exhibit a broader range of variations in terms of shape, intensity, color, and background compared to brain tissue. Results are provided in the Supporting Information (S1 Text and S1 Appendix).

## U-Net series for BTS

U-Net, a semantic segmentation architecture like fully convolutional networks (FCN) [28], is characterized by its U-shaped structure. The 'encoder', located on the left branch, performs down-sampling to extract image features. Each feature map undergoes two 3x3 convolutional layers with ReLU activation, followed by four rounds of maximum pooling. The 'decoder', on the right branch, handles up-sampling using transposed convolution to enlarge feature maps and concatenate them with corresponding features from the encoder, ensuring consistent feature map sizes and producing a segmentation output matching the input image size.

UNet++ (UNet2+) improves upon U-Net in three keyways. It integrates dense skip connections, inspired by DenseNet, to retain all prior feature maps, redesigns the skip path to bridge the semantic gap between the encoder and decoder paths, and introduces deep supervision to adaptively adjust model complexity for optimal computational efficiency and performance. To maintain naming consistency, UNet++ is denoted as UNet2+.

UNet3+ distinguishes itself by employing full-scale skip connections, connecting feature maps of varying scales via max-pooling and bilinear up-sampling. Additionally, it introduces the classification-guided module (CGM) to identify object boundaries, enhancing segmentation accuracy while reducing over-segmentation.

## Proposed adaptive fusion strategy for segmentation results

Image fusion involves merging multiple images into one and integrating their features. Given that different BTS models may yield varying segmentation masks with some inaccuracies, our study aims to minimize these errors and enhance accuracy through result fusion. If both BTS model outputs (mask1, mask2) segment fewer brain pixels across most slices, a union operation ($\cup$) is favored to include more accurately segmented brain pixels. Conversely, if more false alarms are present, an intersection operation ($\cap$) is applied to reduce such errors.

To determine the appropriate fusion operation, we define several terms. '$num_{miss}$' represents the total number of missing brain pixels in the output mask when a BTS model segments less brain tissue in a training dataset. '$num_{overseg}$' indicates the total number of over-segmented pixels in the output mask when the model mistakenly identifies non-brain pixels as brain tissue. The fusion strategy F is then expressed as follows:

$$F = \begin{cases} \text{mask}_1 \cup \text{mask}_2, & \text{if num}_{\text{miss}} \geq \text{num}_{\text{overseg}} \text{ for both models} \\ \text{mask}_1 \cap \text{mask}_2, & \text{if num}_{\text{miss}} < \text{num}_{\text{overseg}} \text{ for both models} \end{cases} \tag{1}$$

Here, we note that Eq. 1 does not account for scenarios where one model yields a higher ' $num_{miss}$ ' count while the other produces more ' $num_{overseg}$ '. In such cases, introducing a third model could help resolve the tie by selecting two models that meet the condition outlined in Eq. 1.

When there are more than four BTS models available, multiple pairs can form fusion strategies like F1, F2, etc. However, having more fusion strategies means training more models, which might not offer significant benefits compared to training a single, more complex model. Based on our experience with U-Net series models, we advise against excessive use of multiple fusion strategies. These models often produce more false positives than missed BTS. Therefore, for this study, we've chosen the intersection operation (Fig 3) as our fusion strategy.

### Evaluation indexes

To compare segmentation performances between different single models and fusion, two metrics were used. One is the Intersection over Union (IoU) defined in Eq. 2, which measures overlap between two masks. Its range is [0–1], with higher values indicating better segmentation performance.

$$IoU = \frac{Overlap\ Pixels}{Union\ Pixels} \tag{2}$$

The other metric is Hausdorff Distance (HD), which is calculated between two contours A and B, each considered as a set of pixels on the boundary.

$$HD(A, B) = \max\left(h(A, B), h(B, A)\right) \tag{3}$$

$$h(A, B) = \max_{a \in A}\left\{\min_{b \in B} d(a, b)\right\} \tag{4}$$

$$d(x, y) = \|x - y\| = \sqrt{\sum_{i=1}^{n} |x_i - y_i|^2} \tag{5}$$

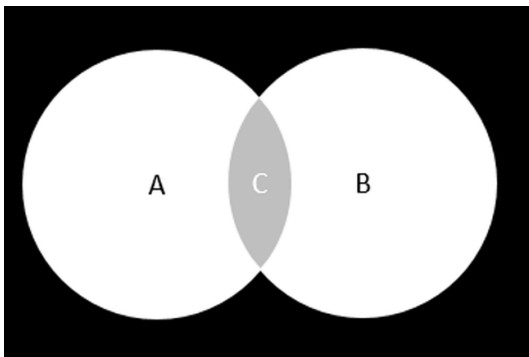

**Fig 3. The intersection operation of two outputs is defined as the overlapped area.** Part C is the fusion result of Image A and Image B.

A lower HD value indicates better segmentation performance. HD is used because IoU is more sensitive to internal contour filling, while HD is more sensitive to contour boundary shape. Solely considering IoU may overlook the prediction accuracy of contour shape.

### Post-processing

The segmentation may not be perfect and may require refinement post-model training. We apply a 9x9 Gaussian filter with unit standard deviation, followed by binarization (threshold = 0.5), to the prediction result to reduce false positives from the original masking operation shown in Fig 4.

### Results

In our study, we analyzed 20 patients, with a gender distribution of 14 males and 6 females, with an average age of 68.9 ± 11.7 years (range: 49–94 years).

Fig 5 illustrates the segmentation outcomes of four sample slices, spanning from the base to the top of the brain. While U-Net demonstrates notable accuracy in delineating brain tissue at the lower cranial regions, it occasionally misclassifies extraneous tissues, particularly in the periorbital area.

UNet2 +, conversely, exhibits limited proficiency in distinguishing tissue at the cranial base, often misidentifying more skull base areas as brain tissue. Nonetheless, it effectively captures all genuine brain tissue regions.

Despite UNet3 + yielding superior results overall, there are instances of unnecessary segmentation. Generally, each U-Net model demonstrates strengths and weaknesses, excelling in segmenting upper brain regions where the ground truth mask typically comprises a single object.

The quantitative assessment of segmentation effectiveness is presented through tabulated IoU and HD metrics, encompassing the entire head from CT image series. Various combinations of optimizers and learning rates (lr) were explored, with the optimal performance of each model summarized in Tables 1 and 2.

In addition to the raw segmentation outputs, the results of post-processing steps were evaluated. Tables 1 and 2 demonstrate the enhancements achieved following the application of a Gaussian filter. Overall, UNet3 + exhibits the highest performance, followed by UNet2+ and U-Net. Notably, an initial learning rate of 0.001 yields superior IoU and lower HD for U-Net and UNet3+ but proves unsuitable for UNet2 + even with adaptive tuning using Adagrad and Adam algorithms.

Upon careful examination of the segmented images from the three models, we found that UNet3 + produced results closest to the ground truth image, yielding higher IoU and lower HD scores. However, the segmentation behavior of

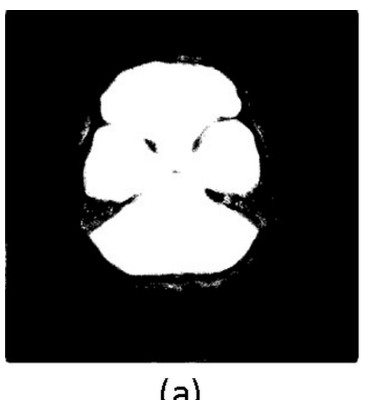 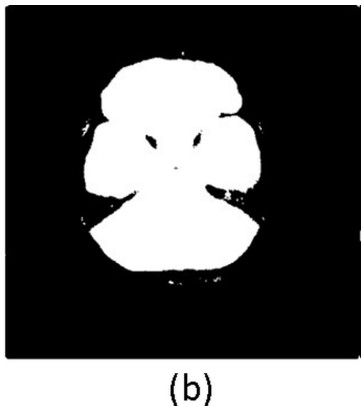

(a) (b)

**Fig 4.** (a) A mask image with some noise near the boundary after model prediction; and (b) the resulting image after Gaussian filtering and binarization of mask in (a).

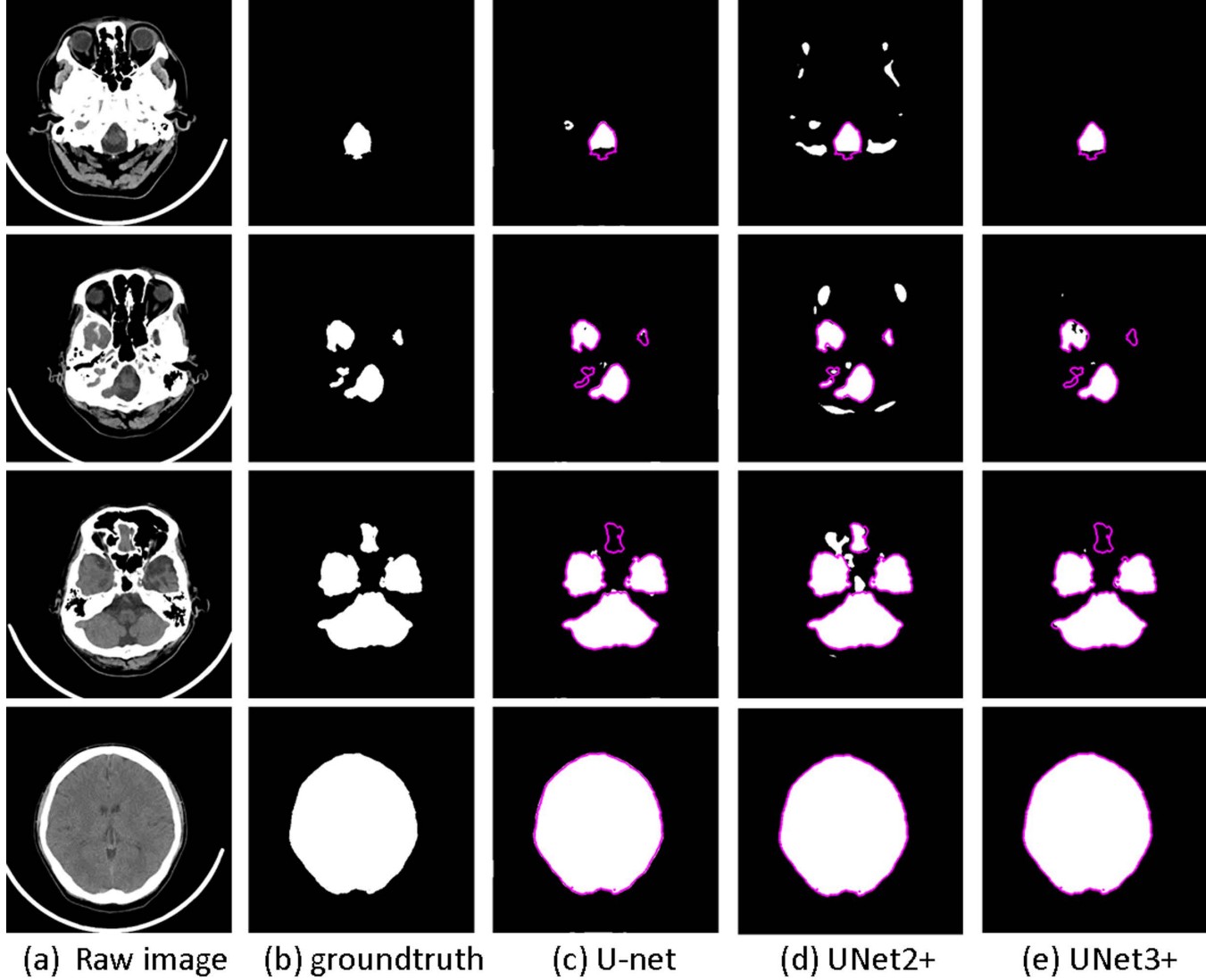

(a) Raw image    (b) groundtruth    (c) U-net    (d) UNet2+    (e) UNet3+

**Fig 5. The results of U-Net, UNet2 +, and UNet3 +.** In (c-e) the ground truth is marked as the contour in magenta to contrast the difference with the segmentation results. (a) Raw image (b) ground truth (c) U-net (d) UNet2+ (e) UNet3+.

**Table 1. IoU and HD of UNets family when the optimizer is set as Adagrad. The best-performing results are highlighted using bold font.**

| Model (initial lr) | Prediction | | Applied filtering | |
|---|---|---|---|---|
| | IoU | HD | IoU | HD |
| U-Net (lr=0.001) | 0.880 | 6.16 | 0.930 | 5.26 |
| UNet2+ (lr=0.0001) | 0.902 | 25.2 | 0.903 | 24.9 |
| UNet3+ (lr=0.0001) | 0.937 | **1.91** | **0.938** | 1.95 |

UNet2+differed notably. Although successfully trained, its results showed significant deviation from the ground truth compared to U-Net and UNet3+.

We selected the best-performing versions of each model based on IoU from Tables 1 and 2. UNet2+was chosen from Table 1, while U-Net and UNet3+were selected from Table 2. Total '$num_{overseg}$' and '$num_{miss}$' counts were then tabulated in Table 3, revealing a consistent trend of $num_{overseg} > num_{miss}$. Consequently, we identified their intersection as a suitable fusion strategy.

We leveraged the diverse segmentation behaviors of UNet2+, U-Net, and UNet3+ by fusing their results. Given UNet2+'s sensitivity to learning rate, the model trained with lr=0.0001 was chosen as the fusion base. The superior models, U-Net and UNet3+, trained with lr=0.00001, were selected for fusion. Table 4 summarizes the best results of each fusion, demonstrating the success of our fusion strategy. Notably, the fusion of UNet2+ and UNet3+achieved the highest IoU (0.955), surpassing the IoU obtained by fusing all three U-Net models (IoU=0.952).

Fusion results (from the same example slices depicted in Fig 5) are illustrated in Fig 6, demonstrating the elimination of almost all excessive areas (false positives) post-fusion.

Fig 7a presents the plotted IoU scores for a patient's entire head CT series, spanning from the bottom to the top of the head. In the legend, the label 'Fusion (best)' refers to the result fusion of UNet2+ and UNet3+. The segmentation performance exhibits greater fluctuation at the bottom (slices 1–7) and top (slices 29–31) of the brain but remains relatively stable in the middle (slices 8–28). The relatively stable region is magnified in Fig 7b. Both plots demonstrate the superior performance and effectiveness of the result fusion strategy compared to the individual UNet-series models.

Additionally, the results obtained from the cat images selected from the COCO dataset are provided in the Supporting Information (S1 Text and S1 Appendix).

**Table 2. IoU and HD of UNets family when the optimizer is set as Adam. lr: learning rate. The best-performing results are highlighted using bold font.**

| Model (initial lr) | Prediction | | Applied filtering | |
|---|---|---|---|---|
| | IoU | HD | IoU | HD |
| U-Net (lr=0.0001) | 0.905 | 4.52 | 0.937 | 4.52 |
| UNet2+ (lr=0.001) | 0.048 | 415.0 | 0.048 | 415.1 |
| UNet3+ (lr=0.001) | 0.947 | **1.29** | **0.948** | 1.69 |

**Table 3. $num_{overseg}$ and $num_{miss}$ of three BTS models. The best-performing results are highlighted using bold font. lr: learning rate.**

| Model (initial lr, optimizer) | Prediction | | Applied filtering | |
|---|---|---|---|---|
| | $num_{overseg}$ | $num_{miss}$ | $num_{overseg}$ | $num_{miss}$ |
| U-Net (lr=0.0001, Adam) | 48134 | 14023 | 48058 | 14031 |
| UNet2+ (lr=0.0001, Adagrad) | 97291 | **13606** | 96438 | **13652** |
| UNet3+ (lr=0.001, Adam) | **25531** | 21636 | **25453** | 21656 |

**Table 4. Fusion results with UNet2+ (optimizer=Adagrad, learning rate=0.0001). The best-performing results are highlighted using bold font.**

| Model (optimizer, learning rate) | Prediction | | Applied filtering | |
|---|---|---|---|---|
| | IoU | HD | IoU | HD |
| U-Net (Adagrad, 0.001) | 0.945 | 1.58 | 0.945 | 2.03 |
| UNet3+ (Adagrad, 0.001) | **0.9550** | **1.33** | **0.9552** | **1.55** |
| Fusion all | 0.952 | 1.61 | 0.952 | 1.61 |

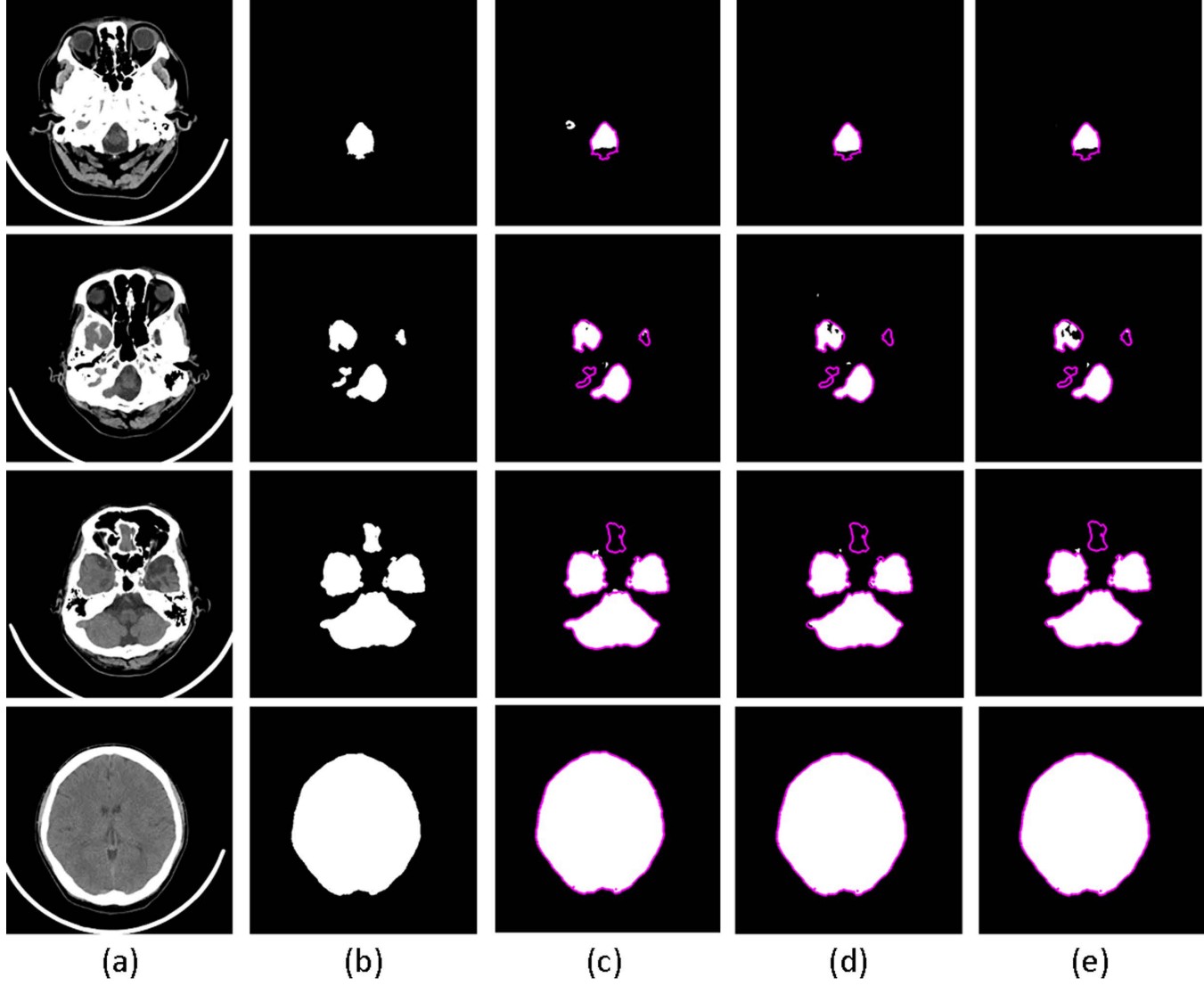

**Fig 6. The results of fusion.** (a) Origin CT image; (b) Ground truth; (c) Fusion of U-Net & UNet2+; (d) Fusion of UNet2+ & UNet3+; (e) Fusion of U-Net & UNet2+ & UNet3+. In (c-e) The ground truth is marked as the contour in magenta to contrast the difference with the prediction results.

## Discussions

According to our findings, the highest IoU achieved for the NCCT dataset is 0.955, with the most favorable HD recorded at 1.33, both obtained through fusion with UNet2+ and UNet3+ architectures (refer to Table 4). Evaluation on the COCO dataset revealed a peak IoU of 0.463 and a minimum HD of 584.1, also achieved through fusion with UNet2+ and UNet3+ (refer to S1 Table). Therefore, in this study, we successfully propose an adaptive results fusion strategy tailored to enhance the accuracy of BTS tasks conducted by multiple DL-based networks.

It is noteworthy that the performance of the fusion of UNet2+ & UNet3+ surpasses that of the fusion of all three U-Net series. The underlying reasons for this disparity remain unknown. However, as depicted in Fig 7, the IoU scores of individual U-Net series CNNs demonstrate suboptimal performance at the bottom slices (1–7) of the head CT scan. Notably, at these bottom slices, the IoU scores follow the order: UNet3+ > U-Net > UNet2+. Conversely, Table 4 highlights UNet3+ as

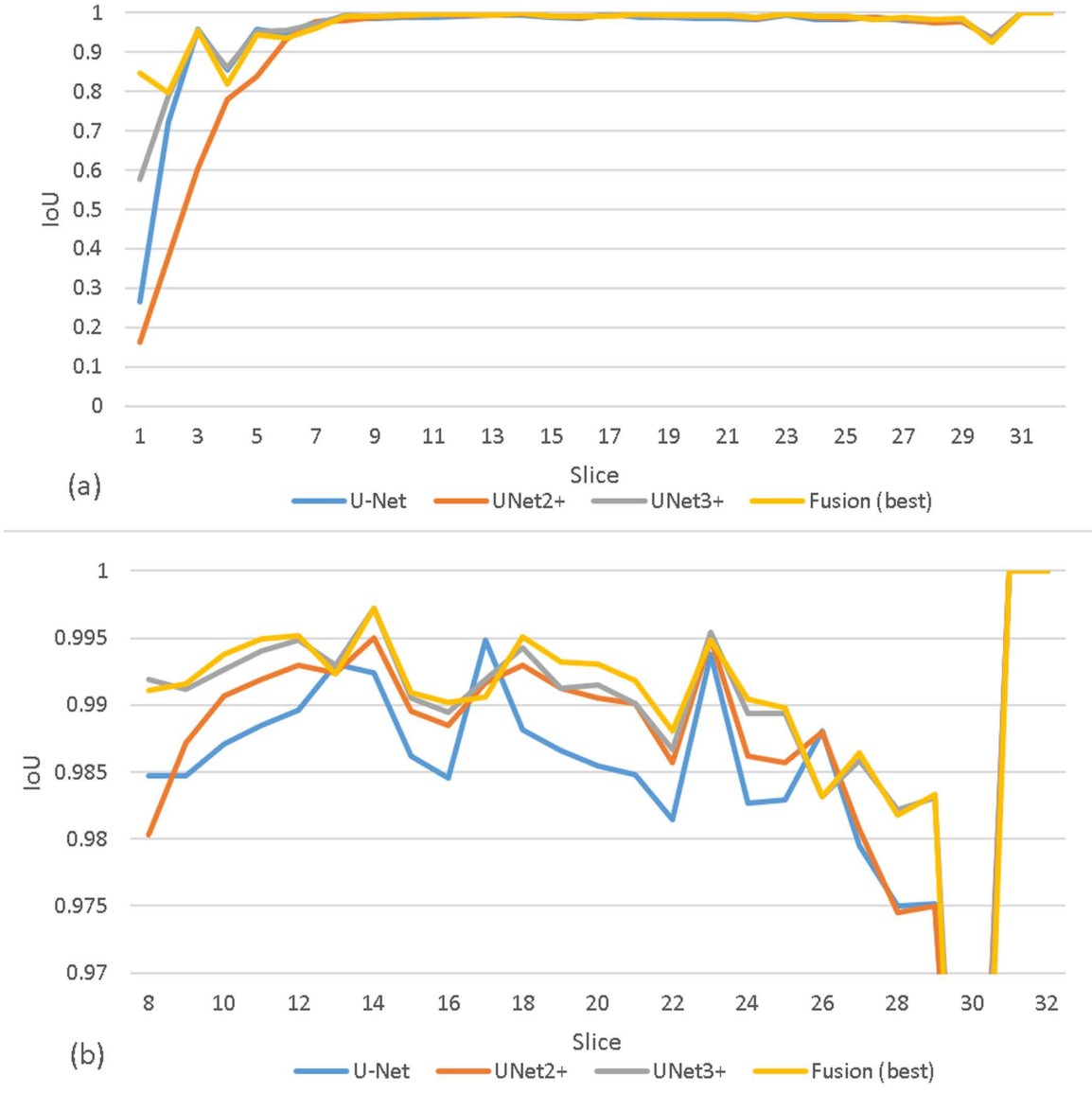

**Fig 7. The IoU score variations among U-Net, UNet2 +, UNet3 +, and the fusion of UNet2 + & UNet3+ across an entire head CT series, from the bottom of the brain to the top of the head.** (a) illustrates this variation; (b) provides a magnified view of IoU variation specifically during slices 8-32 for each model.

achieving the best performance in terms of '$num_{overseg}$', while UNet2 + outperforms in terms of '$num_{miss}$'. Given the observed compensatory effects between the performances of UNet3+ and UNet2 +, it is reasonable to anticipate favorable performance from the fusion of UNet2 + & UNet3 +. However, considering the moderate performance of U-Net, which lies between UNet2+ and UNet3 +, the cumulative effect on performance may not be significant.

Based on the findings presented in d Table 2, it is apparent that the efficacy of our selected filter, a 9x9 Gaussian filter with a unit standard deviation, varies depending on the optimizer used. Specifically, when Adagrad is employed, the filter exhibits enhanced performance, characterized by increased IoU and decreased HD. In contrast, when Adam is utilized

as the optimizer, the filter's performance is less favorable, with only marginal improvements in IoU and no significant reduction, or even an increase, in HD. However, when examining the more complex cat images in the COCO dataset, as presented in Table 5 and Table 6, the filter effects under both Adagrad and Adam optimizers appear limited. This observation suggests that the efficacy of the Gaussian filter, particularly the 9x9 Gaussian filter with a unit standard deviation, may be influenced differently in datasets with varying image complexities. Specifically, while the filter performs well on relatively simple images such as NCCT, its effectiveness may be less pronounced when dealing with more intricate images such as those found in the COCO dataset. Additionally, the choice of optimizer, such as Adagrad or Adam, may further modulate the filter's performance, highlighting the importance of optimization strategies in image segmentation tasks.

The examination of the filter effect, as detailed in Table 3, demonstrates a notable trend: while the application of the filter tends to decrease the occurrence of '$num_{overseg}$', it also correlates with a simultaneous increment in '$num_{miss}$'. This phenomenon resembles the trade-off observed in diagnostic testing, where increasing sensitivity often comes at the expense of decreased specificity. We propose that this principle extends to the application of filter algorithms, where adjustments in parameters may influence the balance between segmentation metrics.

Upon close examination of Table 4, we observed a subtle discrepancy between experiments yielding the highest IoU (0.9552) and the lowest HD (1.33), corresponding to experiments with and without Gaussian filtering, respectively. This variance may be attributed to IoU's focus on mask overlap and HD's emphasis on boundary accuracy. However, even with this slight difference in perspective, it does not undermine the efficacy of fusing UNet2+ and UNet3+ over fusing all three U-Net models.

The results obtained from the cat images selected from the COCO dataset (as detailed in the Supporting Information (S1 Text and S1 Appendix)) were not as favorable as those from NCCT images, primarily due to the greater complexity of the cat images. However, the fusion of results aided in reducing the non-Region of Interest (ROI) area. Specifically, the fusion of UNet2+ and UNet3+ yielded the highest segmentation performance. This finding is consistent with the observations in Table 4 regarding the BTS task.

There are several limitations to this study. Firstly, the dataset used for the BTS task comprised a limited sample size of 20 patients. However, it is noteworthy that the patient cohort encompassed a wide age range (49–94 years old), with a predominantly male distribution (Male/Female 14/6), thus offering a representative diversity among stroke patients. Moreover, U-Net models have demonstrated efficient performance even with smaller training datasets. Hence, despite the small sample size, we consider our results robust, given the pilot nature of the study. Secondly, our study only employed one type of filter, namely the Gaussian filter. It is possible that different types of filters could yield varying impacts on the segmentation

**Table 5. IoU and HD of UNets family when the optimizer is set as Adagrad. The best-performing results are highlighted using bold font.**

| Model (initial lr) | Prediction | | Applied filtering | |
|---|---|---|---|---|
| | IoU | HD | IoU | HD |
| U-Net (lr = 0.001) | 0.381 | 1224.8 | 0.382 | 1279.7 |
| UNet2+ (lr = 0.0001) | 0.391 | 1657.5 | 0.392 | 1678.0 |
| UNet3+ (lr = 0.001) | 0.472 | 684.1 | **0.472** | **667.6** |

**Table 6. IoU and HD of UNets family when the optimizer is set as Adam. The best-performing results are highlighted using bold font.**

| Model (initial lr) | Prediction | | Applied filtering | |
|---|---|---|---|---|
| | IoU | HD | IoU | HD |
| U-Net (lr = 0.001) | 0.374 | 869.8 | 0.374 | 896.3 |
| UNet2+ (lr = 0.001) | 8.39e-09 | 467.9 | 8.39e-09 | 467.9 |
| UNet3+ (lr = 0.0001) | **0.510** | **531.3** | 0.510 | 538.8 |

outcomes. Thirdly, we exclusively utilized the cat series from the COCO dataset. As the COCO dataset contains numerous object classes, the segmentation results may vary across different object categories. However, due to the extensive number of object classes in the COCO dataset, it may not be feasible to validate the performance across all classes.

In summary, result fusion emerges as a compelling solution to address the inherent limitations of individual U-Net models and effectively rectify mis-segmented regions. The incorporation of this fusion strategy into our proposed BTS method not only enhances the reliability of skull stripping but also holds promise in delivering timely and invaluable insights for accurate lesion volume estimation in future stroke diagnosis.

## Conclusions

In our study, we proposed an adaptive results-fusion strategy to reduce false alarms in BTS tasks using multiple DL-based networks. By combining results from different U-Net models, we significantly decreased false alarms, leading to improved segmentation accuracy with higher IoU and lower HD scores.

## Supporting information

**S1 Text. Investigation into the generalization capability of the proposed adaptive fusion strategy.**
(DOCX)

**S1 Appendix. Investigation into the generalization capability of the proposed adaptive fusion strategy.**
(DOCX)

**S1 Table. Result fusion with UNet2+ (optimizer＝Adagrad, learning rate＝0.0001).** The best-performing results are highlighted using bold font.
(DOCX)

**S1 Fig. Example image (left) and corresponding ground truth mask (right).**
(TIF)

**S2 Fig. Illustration of segmentation results.** (a) Original cat image; (b) Ground truth image; (c) Segmentation result using U-Net; (d) Segmentation result using UNet2＋; (e) Segmentation result using UNet3＋; (f) Optimal fusion result (UNet2＋& UNet3+).
(TIF)

## Author contributions

**Conceptualization:** Bang-Chuan Chen, Ren-Hung Hwang, Wei-Min Liu.

**Data curation:** Bang-Chuan Chen, Jyh-Wen Chai, Chi-Hsiang Chou.

**Formal analysis:** Chung-Yi Shen, Wei-Chuan Chiang.

**Resources:** Jyh-Wen Chai.

**Writing – original draft:** Bang-Chuan Chen, Chung-Yi Shen.

**Writing – review & editing:** Bang-Chuan Chen, Chi-Hsiang Chou, Wei-Min Liu.

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
