## [Decision Letter · Decision Letter 0]

2 Jul 2024

Dear Dr. Chou,

Thank you for submitting your manuscript to PLOS ONE. After careful consideration, we feel that it has merit but does not fully meet PLOS ONE’s publication criteria as it currently stands. Therefore, we invite you to submit a revised version of the manuscript that addresses the points raised during the review process.

We look forward to receiving your revised manuscript.

Kind regards,

Xiaohui Zhang

Academic Editor

PLOS ONE

 [The work is supported by Advanced Institute of Manufacturing with High-tech Innovations, National Chung Cheng University, Chiayi, Taiwan and Taichung Veterans General Hospital with grant number TCVGH-1085505C. It is also supported by National Science and Technology Council with grant number 110-2221-E-194 -011.].  

[The work is supported by Advanced Institute of Manufacturing with High-tech Innovations, National Chung Cheng University, Chiayi, Taiwan and Taichung Veterans General Hospital with grant number TCVGH-1085505C. It is also supported by National Science and Technology Council with grant number 110-2221-E-194 -011.]

 [The work is supported by Advanced Institute of Manufacturing with High-tech Innovations, National Chung Cheng University, Chiayi, Taiwan and Taichung Veterans General Hospital with grant number TCVGH-1085505C. It is also supported by National Science and Technology Council with grant number 110-2221-E-194 -011.].

6. We note that you have indicated that there are restrictions to data sharing for this study. PLOS only allows data to be available upon request if there are legal or ethical restrictions on sharing data publicly. For more information on unacceptable data access restrictions, please see http://journals.plos.org/plosone/s/data-availability#loc-unacceptable-data-access-restrictions. 

7. Your ethics statement should only appear in the Methods section of your manuscript. If your ethics statement is written in any section besides the Methods, please move it to the Methods section and delete it from any other section. Please ensure that your ethics statement is included in your manuscript, as the ethics statement entered into the online submission form will not be published alongside your manuscript. 

8. We note that Figure(s) 1, 2, 4, 5, 6, 8 and 9 in your submission contain copyrighted images. All PLOS content is published under the Creative Commons Attribution License (CC BY 4.0), which means that the manuscript, images, and Supporting Information files will be freely available online, and any third party is permitted to access, download, copy, distribute, and use these materials in any way, even commercially, with proper attribution. For more information, see our copyright guidelines: http://journals.plos.org/plosone/s/licenses-and-copyright.

a. You may seek permission from the original copyright holder of Figure(s)  1, 2, 4, 5, 6, 8 and 9  to publish the content specifically under the CC BY 4.0 license. 

Reviewers' comments:

Reviewer's Responses to Questions

**Comments to the Author**

1. Is the manuscript technically sound, and do the data support the conclusions?

Reviewer #1: Partly

Reviewer #2: Yes

2. Has the statistical analysis been performed appropriately and rigorously?

Reviewer #1: No

Reviewer #2: N/A

3. Have the authors made all data underlying the findings in their manuscript fully available?

Reviewer #1: No

Reviewer #2: No

4. Is the manuscript presented in an intelligible fashion and written in standard English?

Reviewer #1: Yes

Reviewer #2: Yes

Reviewer #1: This paper introduces a model fusion strategy to enhance brain tissue segmentation performance using U-Nets. The proposed technique is straightforward to implement and showed improved performance compared to standalone U-Nets. The paper is written in an easy-to-understand manner. Here are some comments for consideration:

Major comments:

• It seems that all the results from each segmentation model were generated through one trial. To improve the validity of the observations, please train the models multiple times and report the overall segmentation results (e.g., mean and standard deviation of evaluation metrics).

• In the Discussion section, you mention “the cumulative effect on performance may not be significant.” Please include the segmentation results of the fusion of U-Net and UNet3++ to substantiate this claim.

• It is suggested that the authors can provide theoretical explanations for the performance differences among the three U-Nets used in the study based on their architectural differences. This can offer valuable insights to readers regarding the preferable U-Net architecture for the brain tissue segmentation task.

• The computational cost of different segmentation strategies should be discussed in the Discussion section.

Minor comments:

• In the Materials and Methods -- Training section, please use the full name of “BCELoss” (Binary Cross Entropy loss).

• In the Materials and Methods – U-Net series for BTS section, please avoid using “left branch” and “right branch” to describe the encoder and decoder of U-Nets. Such ambiguous descriptions can be misleading.

• In the descriptions of Table 5-7, please specify the results are from the COCO cat segmentation task.

• In Figure 1 caption “the over-segmented results by some brain tissue segmentation algorithms”: please specify the names of the brain tissue segmentation algorithms used.

Reviewer #2: This work describes a fusion framework that combines the prediction of multiple UNet models to improve the overall classification accuracy of the model. The following minor comments and questions should be addressed before the paper is ready for publication:

1. This work uses simple intersection as the fusion operation, while more complex, nonlinear forms of fusion exist, such as an element-wise multiplication (of the latent space, not the final prediction). See https://doi.org/10.1016/j.cma.2023.116277 and https://doi.org/10.1007/s10915-022-01881-0 for examples. The authors should discuss whether those nonlinear forms of data fusion have a potential use in the current case.

2. For classification models, logits are typically generated and then passed through a final activation layer to generate the binary classification output. In this case, the authors chose to perform fusion on the activated output, what about the possibility of performing fusion on the logits, then apply a single activation to the fused logits? The authors can maybe discuss a bit along this line.

3. It seems like the authors manually varied some hyper-parameters of the UNet models to investigate their effectiveness. automatic hyperparameter optimization models have been widely explored and applied, such as https://doi.org/10.1002/widm.1484, https://doi.org/10.1007/s00707-023-03691-3. Relevant works should be discussed in the literature review.

4. The authors mentioned "However, having more fusion strategies means training more models, which might not offer significant benefits compared to training a single, more complex model.". This is an interesting point that was not supported by data in the current manuscript. The authors should list the number of trainable parameters and training times for each model used in this work, and compared the fusion results (both in terms of training time and accuracy) with a single UNet model with comparable number of trainable parameters (as the two fused models combined).

**Do you want your identity to be public for this peer review?** For information about this choice, including consent withdrawal, please see our Privacy Policy

Reviewer #1: No

Reviewer #2: No

---

## [Author Response · Author response to Decision Letter 1]

28 Feb 2025

November 27th, 2024

Dr. Xiaohui Zhang

Academic Editor

PLOS ONE

Dear Dr. Zhang,

We are grateful for the opportunity to revise our manuscript entitled "Refining CT Image Analysis: Exploring Adaptive Fusion in U-Nets for Enhanced Brain Tissue Segmentation" (PONE-D-24-17378). We appreciate the insightful comments and suggestions provided by you and the reviewers. We have addressed each point raised and made the necessary revisions to improve our manuscript. Besides, we also claim that ‘The funders provided financial support for the Article Processing Charge (APC) but had no role in study design, data collection and analysis, decision to publish, or preparation of the manuscript’. Below, we provide a detailed response to the editor's and reviewers' comments. Please also noted that in the revised article, we have highlighted the changes made in response to the reviewer’s comments in green, while minor revisions from the previous version of the article are highlighted in yellow.

Editor:

1. Ensure Manuscript Meets Style Requirements:

Comment: "Please ensure that your manuscript meets PLOS ONE's style requirements, including those for file naming. The PLOS ONE style templates can be found at https://journals.plos.org/plosone/s/file?id=wjVg/PLOSOne_formatting_sample_main_body.pdf and https://journals.plos.org/plosone/s/file?id=ba62/PLOSOne_formatting_sample_title_authors_affiliations.pdf."

Response: Thank you for highlighting this requirement. We have carefully revised the manuscript to adhere to PLOS ONE's style requirements, including the formatting and file naming conventions. The updated manuscript now conforms to the provided templates.

2. Code Sharing Guidelines:

Comment: "Please note that PLOS ONE has specific guidelines on code sharing for submissions in which author-generated code underpins the findings in the manuscript. In these cases, we expect all author-generated code to be made available without restrictions upon publication of the work. Please review our guidelines at https://journals.plos.org/plosone/s/materials-and-software-sharing#loc-sharing-code and ensure that your code is shared in a way that follows best practice and facilitates reproducibility and reuse."

Response: The codes can be found in https://github.com/limrs2011/PLOSone2024/

3. Funding Information:

Comment: "We note that the grant information you provided in the ‘Funding Information’ and ‘Financial Disclosure’ sections do not match. When you resubmit, please ensure that you provide the correct grant numbers for the awards you received for your study in the ‘Funding Information’ section."

Response: The discrepancies in the grant information have been resolved. The correct grant numbers are now provided consistently in the ‘Funding Information’ section.

4. Role of Funders:

Comment: "Please state what role the funders took in the study. If the funders had no role, please state: 'The funders had no role in study design, data collection and analysis, decision to publish, or preparation of the manuscript.' If this statement is not correct you must amend it as needed. Please include this amended Role of Funder statement in your cover letter; we will change the online submission form on your behalf."

Response: We have included the statement “The funders provided financial support for the Article Processing Charge (APC) but had no role in study design, data collection and analysis, decision to publish, or preparation of the manuscript.” in the cover letter and manuscript.

5. Funding Information Removal from Acknowledgments:

Comment: "We note that you have provided funding information that is not currently declared in your Funding Statement. However, funding information should not appear in the Acknowledgments section or other areas of your manuscript. We will only publish funding information present in the Funding Statement section of the online submission form. Please remove any funding-related text from the manuscript and let us know how you would like to update your Funding Statement."

Response: Funding-related text has been removed from the Acknowledgments section and included in the ‘Funding Information’ section.

6. Data Sharing Restrictions:

Comment: "We note that you have indicated that there are restrictions to data sharing for this study. PLOS only allows data to be available upon request if there are legal or ethical restrictions on sharing data publicly. For more information on unacceptable data access restrictions, please see http://journals.plos.org/plosone/s/data-availability#loc-unacceptable-data-access-restrictions. Before we proceed with your manuscript, please address the following prompts:

b) If there are no restrictions, please upload the minimal anonymized data set necessary to replicate your study findings to a stable, public repository and provide us with the relevant URLs, DOIs, or accession numbers. For a list of recommended repositories, please see https://journals.plos.org/plosone/s/recommended-repositories. You also have the option of uploading the data as Supporting Information files, but we would recommend depositing data directly to a data repository if possible."

Response: Thank you for your comments. We would like to address the data sharing restrictions for our study as follows:

a) There are ethical restrictions on sharing a de-identified data set because we cannot ensure that publicly shared brain CT images will not be reconstructed into three-dimensional images, potentially revealing identifiable facial features of the patients and thus posing a privacy risk. Due to these concerns, we did not include data sharing in our initial IRB application. As a result, the current data are not suitable for public sharing. These restrictions have been imposed by our Institutional Review Board (IRB) to protect patient confidentiality and privacy.

b) As the data contain sensitive information and there are ethical restrictions in place, we are unable to upload the anonymized data set to a public repository.

We hope this explanation clarifies the restrictions. If there are any further questions or if you require additional information, please do not hesitate to contact us.

7. Ethics Statement Location:

Comment: "Your ethics statement should only appear in the Methods section of your manuscript. If your ethics statement is written in any section besides the Methods, please move it to the Methods section and delete it from any other section. Please ensure that your ethics statement is included in your manuscript, as the ethics statement entered into the online submission form will not be published alongside your manuscript."

Response: The ethics statement has been moved to the Methods section and removed from any other sections.

8. Copyrighted Images:

Comment: "We note that Figure(s) 1, 2, 4, 5, 6, 8, and 9 in your submission contain copyrighted images. All PLOS content is published under the Creative Commons Attribution License (CC BY 4.0), which means that the manuscript, images, and Supporting Information files will be freely available online, and any third party is permitted to access, download, copy, distribute, and use these materials in any way, even commercially, with proper attribution. For more information, see our copyright guidelines: http://journals.plos.org/plosone/s/licenses-and-copyright. We require you to either (1) present written permission from the copyright holder to publish these figures specifically under the CC BY 4.0 license, or (2) remove the figures from your submission:

a. You may seek permission from the original copyright holder of Figure(s) 1, 2, 4, 5, 6, 8 and 9 to publish the content specifically under the CC BY 4.0 license.

Response: Thank you for your comments regarding the figures in our submission. We would like to clarify the sources of the figures mentioned and ensure compliance with the CC BY 4.0 license:

Figures 1, 2, 4, 5, and 6: These figures are brain CT images obtained from our own hospital. These images are original and not subject to third-party copyright restrictions. Therefore, we confirm that these images can be published under the CC BY 4.0 license without needing additional permissions. The figure captions will include the following text: "© 2024 [Taichung Veterans General Hospital]. Published under the Creative Commons Attribution License (CC BY 4.0)."

Figures 8 and 9: These figures were obtained from a publicly available dataset [27] with license CC BY-SA 4.0. The figure captions will include the text: "Reprinted from [27] under CC BY-SA 4.0 license." We hope this clarifies the situation. Please let us know if you require any further information or have any additional instructions for us.

Reviewer #1:

1. Train Models Multiple Times:

Comment: "It seems that all the results from each segmentation model were generated through one trial. To improve the validity of the observations, please train the models multiple times and report the overall segmentation results (e.g., mean and standard deviation of evaluation metrics)."

Response: Thank you for the suggestions. To further enhance the validity of our observations and prevent data overfitting, we have included an additional public dataset with a larger number of patients in the revised manuscript. We also conducted 5-fold cross-validation experiments to verify our previous results obtained from the private dataset. The mean and standard deviation of the evaluation metrics are presented in Table 2. Additionally, the previous Tables 1 and 2 in the earlier version have been combined into a new Table 1 to simplify the presentation of results.

2. Include Segmentation Results of U-Net and UNet3++ Fusion:

Comment: "In the Discussion section, you mention 'the cumulative effect on performance may not be significant.' Please include the segmentation results of the fusion of U-Net and UNet3++ to substantiate this claim."

Response: We apologize for the confusion caused by the sentence. Although we have shown that appropriate fusion of the results of two model could enhance the segmentation performance, we cannot pursue the best result by unlimited fusion. The ‘cumulative effect’ here means fusing the results from more than two models. That is why in Table 4 the IoU of ‘fusion all’ is not as high as fusion of UNet2+ and UNet3+. The descriptions for clarification have been added in L399~L407.

Since the design of Table 4 is to set UNet2+ as the fusion base, we did not include the fusion result of U-Net and UNet3+, otherwise it literally becomes the result of ‘fusion all’. In the following table we followed the reviewer’s suggestion and used ISLES2018 as the dataset to show the results of single models and all combinations of fusion. From the last three rows we can see that all fusions bring slightly better segmentation than the single models in this dataset, but we do not guarantee the trend remains forever. Different segmentation models, different datasets, or different segmentation tasks (other than BTS) may produce different results. Validation of this will take much longer time and computation resource. We can only humbly share our findings on limited models and datasets in the current manuscript.

Model Prediction

IoU HD

U-Net 0.956±0.004 0.608±0.143

UNet2+ 0.955±0.002 0.717±0.251

UNet3+ 0.957±0.005 0.491±0.100

UNet & UNet2+ 0.9593 0.400

UNet & UNet3+ 0.9602 0.381

UNet2+ & UNet3+ 0.9587 0.346

3. Theoretical Explanations for Performance Differences:

Comment: "It is suggested that the authors can provide theoretical explanations for the performance differences among the three U-Nets used in the study based on their architectural differences. This can offer valuable insights to readers regarding the preferable U-Net architecture for the brain tissue segmentation task."

Response: Thank reviewer for the suggestions. In addition to the original description of U-Net series networks in section 2.3, we have added more description in L386~L391,

4. Discuss Computational Cost:

Comment: "The computational cost of different segmentation strategies should be discussed in the Discussion section."

Response: Thank reviewer for the suggestions. We have added description to address the computation cost issue in L399~L407.

Minor Comments:

5. Full Name of BCELoss:

Comment: "In the Materials and Methods -- Training section, please use the full name of ‘BCELoss’ (Binary Cross Entropy loss)."

Response: The full name “Binary Cross Entropy loss” is now used in the Training section.(L168)

6. Avoid Ambiguous Descriptions:

Comment: "In the Materials and Methods – U-Net series for BTS section, please avoid using ‘left branch’ and ‘right branch’ to describe the encoder and decoder of U-Nets. Such ambiguous descriptions can be misleading."

Response: Descriptions such as “left branch” and “right branch” have been removed to avoid ambiguity.(L137&L140)

7. Specify Results from COCO Cat Segmentation Task:

Comment: "In the descriptions of Table 5-7, please specify the results are from the COCO cat segmentation task."

Response: The descriptions of Tables 5-7 now specify that the results are from the COCO cat segmentation task.

8. Specify Brain Tissue Segmentation Algorithms:

Comment: "In Figure 1 caption ‘the over-segmented results by some brain tissue segmentation algorithms’: please specify the names of the brain tissue segmentation algorithms used."

Response: The names of the brain tissue segmentation algorithms used are specified in on Figure 1 (c) and (f).

Reviewer #2:

1. Discuss Nonlinear Forms of Data Fusion:

Comment: "This work uses simple intersection as the fusion operation, while more complex, nonlinear forms of fusion exist, such as an element-wise multiplication (of the latent space, not the final prediction). See https://doi.org/10.1016/j.cma.2023.116277 and https://doi.org/10.1007/s10915-022-01881-0 for examples. The authors should discuss whether those nonlinear forms of data fusion have a potential use in the current case."

Response: Thank you for your comment. A discussion on potential uses of nonlinear forms of data fusion, such as element-wise multiplication, has been included. We have also referenced the suggested works to support this discussion.

---

## [Decision Letter · Decision Letter 1]

13 Apr 2025

Refining CT image analysis: Exploring adaptive fusion in U-Nets for enhanced brain tissue segmentation

PONE-D-24-17378R1

Dear Dr. Chou,

We’re pleased to inform you that your manuscript has been judged scientifically suitable for publication and will be formally accepted for publication once it meets all outstanding technical requirements.

Kind regards,

Xiaohui Zhang

Academic Editor

PLOS ONE

Additional Editor Comments (optional):

Reviewers' comments:

Reviewer's Responses to Questions

**Comments to the Author**

Reviewer #1: All comments have been addressed

Reviewer #2: All comments have been addressed

2. Is the manuscript technically sound, and do the data support the conclusions?

Reviewer #1: Yes

Reviewer #2: Yes

3. Has the statistical analysis been performed appropriately and rigorously?

Reviewer #1: Yes

Reviewer #2: Yes

4. Have the authors made all data underlying the findings in their manuscript fully available?

Reviewer #1: No

Reviewer #2: Yes

5. Is the manuscript presented in an intelligible fashion and written in standard English?

Reviewer #1: Yes

Reviewer #2: Yes

Reviewer #1: The authors have addressed all the previous comments. I do not have additional comments. The manuscript is ready for the next stage.

Reviewer #2: The authors have sufficiently addressed the reviewer comments, and the manuscript can be considered for publication.

**Do you want your identity to be public for this peer review?** For information about this choice, including consent withdrawal, please see our Privacy Policy

Reviewer #1: No

Reviewer #2: No

---

## [Editor Report · Acceptance letter]

PONE-D-24-17378R1

PLOS ONE

Dear Dr. Chou,

I'm pleased to inform you that your manuscript has been deemed suitable for publication in PLOS ONE. Congratulations! Your manuscript is now being handed over to our production team.

Kind regards,

on behalf of

Dr. Xiaohui Zhang

Academic Editor

PLOS ONE